

# Singular variations in geomagnetic disturbance content at auroral latitudes

Abraham Abraham[1, 2], Gangadharan Renuka[2], Cherian Ligi[1], Joseph Mathew Tiju[1, 2], Anie

5  Varghese Asha[3], S. B. Chandran Rakesh[2, 4], Varghese Blessy[1].

1. Department of Physics, Christian College, Chengannur, Kerala, India.

2. Department of Physics, University of Kerala, Trivandrum, Kerala, India.

3. Department of Physics, SH College, Thevara, Kochi, Kerala, India.

4. Department of Physics, SD College, Alleppey, Kerala, India.

Corresponding Author: Dr Abraham A.

Email: isureshabraham@gmail.com, Ph: 91-9447977385

## Abstract

The geomagnetic field consists of temporal variations induced primarily by the variations in the interplanetary magnetic field. This paper attempts to develop the latitude profile of fluctuations in the geomagnetic field for 34 observatories across the Earth during solar cycle 23 (1997-2008). It is noticed that the disturbance content is low for

20  equatorial stations and high for auroral stations. We show that the latitude profile of disturbance exhibits 'Knee' behaviour, with the fluctuation content rising sharply beyond this critical latitude. The threshold latitude beyond which the stations are subject to high geomagnetic disturbance and hence become increasingly susceptible to associated geo-electric hazard is precisely determined near 52° latitude. The pattern of increasing geomagnetic fluctuations however ends around the auroral oval beyond where singular variations are observed.

25



# 1. Introduction

Magnetic fields pervade the entire universe. The Earth has its own magnetic field called the geomagnetic field. The geomagnetic field varies over time and space. The study of these variations facilitates a better understanding of the mechanisms that cause them. The Earth's magnetic field can be approximated to a dipole embedded within the earth (Gilbert, 1600). The improvement made by Gauss in 1839 on the dipolar model of geomagnetism was a quantum leap from status quo considering the fact that the mathematical base he introduced in the 19[th] century holds even today in spite of later

advancements in theories of origin of geomagnetism including the dynamo theory. Besides its spatial variations, the geomagnetic field varies at different time scales ranging from seconds to millions of years (Constable and Constable, 2004). The long term variations are due to the changes in the dynamo region within the Earth. The short duration fluctuations are due to the current systems in the atmosphere and the magnetosphere (Courtillot and Mouel, 1988). This paper explores short term variations in the geomagnetic field in the light of the specific consequences of such fluctuations on technological

systems.

       Geomagnetic disturbances induce currents in technological systems such as electric power transmission grids, telecommunication cables, oil and gas pipe lines and signalling systems (Boteler et al., 1998) and are referred to as geomagnetically induced current (GIC)'s. Kappenman (2003) has reported that damage sustained by electric power grids due to induced currents resulting from geomagnetic disturbances first affect the auroral regions and extend up to the mid-

latitudes and low-latitudes.

## 2. Methodology

       The amount of fluctuations in the geomagnetic field is gauged by employing the daily Vertical Variance (VV) method. The VV is the modified general and simpler form of the VV index reported by Abraham et al. (2010), which followed from the attempts to precisely determine the degree of disturbance in various time series data sets of geomagnetic

field and IMF. The linear relationship between disturbance content in geomagnetic field and IMF was established therein.





The VV is intended to assign a quantifying numerical value to fluctuations in data sets which are time dependent functions varying within specified temporal end points.

The VV for any data set consisting of values $y(x)$, $y_{i+1}$ and $y_i$ being adjacent data values of $y$ corresponding to adjacent time values $x_{i+1}$ and $x_i$;

$$VV = \sqrt{\frac{\sum_{i=0}^{n-1}|y_{i+1} - y_i|^2}{n}}, \qquad (1)$$

$$\text{where, } n = \frac{(X_B - X_A)}{(x_{i+1} - x_i)}.$$

$X_B$ and $X_A$ are the extreme time limits of the period for which the VV is being determined. The VV is the root mean square of the variation in the dependent variable in the data under assessment. The values of the independent variable ($x$)

10 are not involved in the computation of the VV and its utility is limited to determining the number of samples of the dependent variable over which the averaging is performed in Eq. 1. The VV is used in this paper in place of the VV index used in Abraham et al. (2010) so that mixing of units is avoided and universality is maintained.

While determining the daily VV of geomagnetic data, 24 hour temporal limits are applied to minute data obtain from WDC, Kyoto. In this instance, the VV is essentially the root mean square of the daily variation in the horizontal

15 component of geomagnetic field, B. The expression for determining VV is thus modified.

$$VV = \sqrt{\frac{\sum_{i=0}^{1439}|B_{i+1} - B_i|^2}{1440}} \qquad (2)$$

The VV quantifies the amount of temporal variations in any time series. The VV being of daily nature, aberrations due to quiet-time $S_q$ variations are minimal.





## 3. Analysis

The averages of the daily VV values of various geomagnetic stations for the years from 1997 through 2008 are analysed here to determine the extent of geomagnetic disturbance characteristic of these stations. These estimates serve as

measures of geomagnetic stress endured by the analysed stations. This analysis also leads to a very definite latitude profile of the disturbance for various geomagnetic stations. The threshold latitude above which the stations tend to be more disturbed and are subject to enhanced Geomagnetic Induced current (GIC) risk is precisely estimated from this profile.

The absolute geomagnetic field data for the stations given in Table 1 has been accessed from World Data Centre (WDC), Kyoto. The VV values of the horizontal component of the geomagnetic field (H) for the stations are determined for

each UT day using 1-minute observations. Computer programs are utilized for calculating the VV values. The VV value for each day is determined from 1440 data elements of H.

In this work, the 34 geomagnetic stations given in Table 1 listed in the order of decreasing absolute geomagnetic (GM) latitude are categorized on the basis of their geomagnetic disturbance as estimated using their mean VV values. The table gives the list of analysed stations along with their ABB codes, geographic latitudes, geographic longitudes,

geomagnetic latitudes, and the absolute geomagnetic latitudes. The table summarises the analysis of data pertaining to the 12 years in solar cycle 23 (1997-2008).

In terms of the GM latitude, 9 stations belong to the low-latitudes, 16 to the mid latitudes and 9 to the high latitudes. The VV values of the horizontal component of the geomagnetic field are determined for all the 4383 days of the years 1997 - 2008 for all the above stations. These mean values are used to categorize the geomagnetic stations.

### 3.1    Mean value analysis and validation (1997-2008)

Monthly, yearly and solar cycle means of the VV values for these stations are determined. Yearly and the solar cycle (23) means of the VV values are given in Table 2. Blanks in the table correspond to non-availability of data. The distinct mean of VV value of each station is characteristic and is considered as the disturbance marker for that station.



Referring to Table 2, two patterns are clearly visible in the results. One, for most stations, progressing from 1997 through 2008, there is distinct peak in the mean VV values during the year 2003. This is particularly true for high-latitude and mid-latitude stations. This pattern is broken for low latitude stations. Year 2003 has been one of exceptional geomagnetic disturbance as evidenced by the large **aa** index, solar wind speed and IMF values (Zerbo et al., 2013).

Simultaneous peaking of conventional geomagnetic disturbance markers and the VV mean values validate the applicability of VV as a disturbance marker.

Further, correlation between the solar wind and the VV values are examined in the pattern of year-wise variations. Stations are selected and grouped into categories such as post-auroral (2 stations, Resolute Bay and Cambridge Bay), pre-auroral (3 stations, Iqaluit, Mawson and Sodankyla), mid-latitude (3 stations, Ottawa, Niemegk and Crozet), and low-latitude

(2 stations, Bangui and Ascension Island) on the basis of their GM latitudes. The averages of the yearly VV values of these four groups of stations are shown in Table 3. The solar wind speed data obtained from the ACE data center are given in Table 4. Group averages of the yearly VV values of the four different categories of stations in Table 3 and the solar wind speed in Table 4 normalized to their respective maxima are plotted against the corresponding years in Figure 1. It is evident from the figure that the averages of the yearly VV values for most station groups (post-auroral, pre-auroral and mid-latitude)

strongly follow the solar wind velocity variations. Only the average VV for the low-latitude group of stations deviate from the solar wind pattern.

Pearson correlation attempted between the average VV of the different station groups and the solar wind is obtained in Table 5. The disturbances in the post-auroral stations and the pre-auroral stations have correlation coefficients of 0.96 and 0.91 with the solar wind speed respectively. Solar wind being an interplanetary phenomenon and the geomagnetic

disturbance being a purely terrestrial measurement, the near perfect correlations obtained point to very strong coupling between these two widely separated phenomena. Solar wind is hence established as the primary driver of geomagnetic disturbance. The VV is also validated as a faithful quantification of geomagnetic disturbance. The open magnetic field topology in the polar region must be primarily responsible for the highest coupling exhibited by the post-auroral stations.

The coupling between geomagnetic disturbance and solar wind holds strong in the mid-latitudes also as

demonstrated by the correlation coefficient value of 0.59. However, the said correlation weakens to a very negligible value

of 0.06 in the low latitudes indicating that the solar wind does not exert significant control on the geomagnetic fluctuations in the equatorial region.

A second pattern of variation is also evident from the VV analysis on the basis of Table 2. Even as there are some erratic variations in-between, there is a sharp fall in the mean VV values (yearly and cycle averages) starting from the auroral

oval (Iqaluit station at Abs GM latitude of 73.25˚) to the equatorial stations (Ascension Island station at Abs GM latitude of 2.74˚)**.** Iqaluit station exhibits the highest mean VV value indicating the highest disturbance for any year (mean VV=577.90, year 2003) and also for the entire solar cycle (mean VV=361.52). Ascension Island station exhibits the lowest mean VV value indicating the least disturbance for any year (mean VV=9.17, year 2008). The solar cycle mean VV of Ascension Island station is also very low (mean VV=20.17), just a 1/18[th] of the value for Iqaluit station. The least disturbed station for

the entire solar cycle is Kakadu (mean VV=17.65) which has a GM latitude of -21.34. Thus the pole-ward stations are markedly more disturbed than the equator-ward as expected owing to the weak solar wind coupling at low latitudes. Further observations on the variation of geomagnetic filed with latitude are detailed in sections 3.2 and 3.3.

## 3.2      Spatial disturbance marking

It is evident from the mean value analysis that high latitude stations have a distinctly high value of VV over and above the low latitude stations clearly indicating a relatively disturbed GM field. Each station has a very distinct VV disturbance marker values during the entire solar cycle and through the analysed years. The Iqaluit station has already been noted as the most disturbed one. Considering the entire solar cycle, the Kakadu station which has a GM latitude of -21.34˚ has been found to be least disturbed.  The most equator-ward station of Ascension Island also has a very small disturbance

value of 20.17. These figures point to a coupling between the IMF and GM field, strongest at high latitudes and weakening towards the low latitudes.

Beyond this general observation, each GM station is disturbed to different extents, as indicated by differing VV values. A disturbance marker intends a pictographic representation of the level of GM disturbance borne by each station. Figure 2 shows the location of the 34 GM stations analysed in Section 3.1 above at their respective positions on the world

map. Each station is shown in the form of a dot, the colour of which is indicative of the mean VV value and thereby the level



of GM disturbance endured by that station as shown in the last column of Table 2. The VV value corresponding to the colour of the dot is shown in the colour bar given alongside the map.

Figure 2 shows the disturbance markers of the analysed GM stations for the solar cycle 23 through the years from 1997 to 2008. The colour code varies from deep blue through dark brown as the normalised VV varies from 0 through 1. The

dark brown dot of Iqaluit station is indicative of it being most disturbed while the deep blue dot of Kakadu station shows it to be least disturbed. All other stations range between in terms of colour in accordance with disturbance as shown by the colour bar on the right edge of the figure. Greenish blue dots like that of Resolute bay show that the station is moderately disturbed. Light blue dots for stations like that of Ottawa and Port Aux Francais show the stations to be lightly disturbed. Red dots for stations like Yellowknife and Mawson show these stations to be highly disturbed. These markers facilitate easy

estimation of GM disturbance of each analysed station on a spatial plane.

## 3.3     Estimation of threshold latitude

It is observed from the station disturbance markers in Figure 2 that stations with low values of GM latitude have low average VV values while those with high values of GM latitude have high average VV values. A progressive increase in disturbance from the low-latitude stations to the auroral stations is thus evident. However, it must be mentioned that regions

considered previously low at risk of GIC's have now been warned of extreme geo-electric events (Torta et al., 2012). Hence, constant reassessment is required as for the spatial profiles of such hazards.

The average yearly geomagnetic VV values are used as estimates of the geomagnetic disturbance for the analysed stations. The VV averages from Table 2 for year 2000 (column 7) and the entire solar cycle (last column) for the 34 geomagnetic stations are used in the analysis. The estimation of the spatial extend of geomagnetic disturbance is repeated

here, once for a particular year (2000) and then for the entire solar cycle (23) to ascertain the findings.

A plot of the year (2000) average of the VV values against the absolute value of GM latitudes of the stations is shown in Figure 3. The plot as observed is very remarkable in nature. The stations have low VV values from equatorial stations to those around 50˚ GM latitude. Beyond this latitude, it is observed that the stations exhibit a marked increase in the mean VV values.



Trend line analysis is used to determine the stations which are more prone to geomagnetic disturbance and associated geo-electric hazard. In view of the abrupt changes in the mean VV values, the stations are grouped into quiet category (VV values up to 50), the pre-auroral category (stations with VV values from 50 to the peak value, Iqaluit station) and the post-auroral category (stations with GM latitudes greater than that of Iqaluit). Linear best fits are attempted for these

categories of stations marked separately in green (quiet), red (pre-auroral) and blue (post-auroral) colours against their absolute GM latitude.

Separate line fit equations are obtained for the quiet and pre-auroral categories of stations as follows.

$$y = 0.087\,x + 30.18 \qquad \text{(Quiet)} \qquad \textbf{(3)}$$

$$y = 13.81x - 656.8 \qquad \text{(Pre-auroral)} \qquad \textbf{(4)}$$

On determining the intersection of these two linear best fits (between quiet and pre-auroral stations), the threshold latitude which offsets the geomagnetic disturbance is located. Solving equations (3) and (4), this threshold GM latitude is obtained as 50.06° using data for the year 2000.

Similar analysis is performed using entire data for solar cycle 23 (1997-2008). The plot of the mean VV values against the absolute GM latitude using the data is shown in Figure 4. The nature of the plots so obtained is very much similar

to that for 2000. Linear fit analysis is performed on the quiet, pre-auroral and post-auroral categories of stations. The line fit equations are obtained for the quiet and pre-auroral categories of stations as follows.

$$y = 0.152\,x + 19.89 \qquad \text{(Quiet)} \qquad \textbf{(5)}$$

$$y = 15.76x - 797.1 \qquad \text{(Pre-auroral)} \qquad \textbf{(6)}$$

On determining the intersection of the two linear best fits, the threshold latitude is located. Solving equations (5)

and (6), the threshold GM latitude is obtained as 52.34° using data for the solar cycle 23. This value is in close agreement with the value of 50.06° obtained for the year 2000.

## 4.      Results and discussion

The disturbance in the IMF may be treated as the common antecedent factor of disturbance for all geomagnetic stations. Vokhmyanin and Ponyavin (2012) have established the close association between the observed geomagnetic fields



and the IMF. The value of the geomagnetic field VV resulting from this analysis is a measure of the fluctuations, the disturbance to which a geomagnetic station in subject.

Two remarkable features of the latitude profile of geomagnetic disturbance is brought forth and explored in this analysis. First, there is a sharp increase in the disturbance content at the threshold or 'knee' latitude. Secondly, the

disturbance content peaks at the auroral oval with singular changes beyond. The magnetic field lines have a trapping geometry in the vicinity of the auroral oval where there is a concentration of such lines owing to the dipolar nature of the geomagnetic field (Milan, 2007). This accentuates the effect of any fluctuation in the solar wind and the IMF causing extreme geomagnetic disturbance near the auroral oval. Referring to Figure 3 and Figure 4, this causes a peak in the VV values in the case of stations such as Iqaluit (mean VV=361.52, Abs GM latitude=73.25˚). Pole-ward of the auroral oval, the

closed dipolar field topology changes to an open field one. The abrupt change in the nature of the field causes singular changes in the level of disturbance just beyond the auroral oval in the direction of the geomagnetic poles. It can be noted in Figure 3 and Figure 4 that disturbance levels as manifested in the VV values change vertically in this region (around 73˚ Abs GM latitude) for the slightest change in latitude. Equator-ward of the auroral oval, a decreased density of field lines characteristic of a dipolar topology weakens the magnetic disturbance in that direction.

Kappenman (2005) has stressed the role of disturbances in geomagnetic field in causing the extreme GIC's during the October 2003 storm leading to power transformer failures. Pirjola (1983), Baker et al. (2004) and Toth et al. (2014) have established the role of geomagnetic fluctuations in initiating and sustaining GIC's. Estimating the geomagnetic disturbance thus serves to obtain a projection of the geo-electric hazards. The significance of estimating geomagnetic disturbance at various locations is thereby underlined.

Pulkkinen et al. (2012) have derived geo-electric field profiles as a function of geomagnetic latitudes. It has been noticed that geo-electric field magnitudes drop significantly across a GM latitude boundary of 40˚- 60˚. This threshold at which the geo-electric field magnitudes drop considerably has been identified as 50˚. Below this threshold GM latitude, the induced geo-electric field magnitudes are about an order of magnitude smaller than those above it.

Even as Pulkkinen et al. (2012) have mentioned that the GIC related geo-electric scenarios drop sharply at latitude boundary sector between 40˚- 60˚ and arrived at a value of 50˚ GM latitude for such decline, they have suggested further confirmation for this finding.

This work utilises the VV values to gauge geomagnetic disturbance and confirms the specific extend of disturbance
to which each geomagnetic station is subject. The daily, monthly and yearly disturbance level of each station is determined from the VV values, the yearly and solar cycle averages published herein. It is evident that stations at higher latitudes are more disturbed than those at low latitudes. There is a threshold GM latitude at which the disturbance level begins to rise sharply exhibiting 'knee' behaviour. This threshold latitude has been determined to be at 50.06˚ for the year 2000 and 52.34˚ for the solar cycle 23, the two results being in close agreement. It can thus be concluded positively that at GM latitudes
above 52˚ the geomagnetic field is more disturbed, these disturbances leading to increased geo-electric risk.

## Acknowledgements

The solar wind velocity data has been obtained from the ACE data center. The geomagnetic field data for the same period is available at http://wdc.kugi.kyoto-u.ac.jp/caplot/index.html provided by World Data Centre (WDC), Kyoto. The
ACE data centre provided the solar wind velocity.  The open source world map outline has been obtained from http://www.fabiovisentin.com/world_map/political_world_map.jpg.

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



# Tables

| Sl No. | Station | ABB Code | GG | GG | GM | GM lat (Abs) |
|---|---|---|---|---|---|---|
| 1 | Alert | ALE | 82.50 | 297.65 | 87.38 | 87.38 |
| 2 | Thule/Qanaq | THL | 77.48 | 290.83 | 87.02 | 87.02 |
| 3 | Resolute Bay | RES | 74.69 | 265.11 | 82.61 | 82.61 |
| 4 | Cambridge Bay | CBB | 69.12 | 254.97 | 76.21 | 76.21 |
| 5 | Hornsund | HRN | 77.00 | 15.55 | 74.02 | 74.02 |
| 6 | Dumont d'Urville | DRV | -66.67 | 140.01 | -73.87 | 73.87 |
| 7 | Iqaluit | IQA | 63.75 | 291.48 | 73.25 | 73.25 |
| 8 | Mawson | MAW | -67.60 | 62.88 | -73.04 | 73.04 |
| 9 | Baker Lake | BLC | 64.32 | 263.99 | 72.66 | 72.66 |
| 10 | Yellowknife | YKC | 62.48 | 245.52 | 68.62 | 68.62 |
| 11 | Abisko | ABK | 68.36 | 18.82 | 66.04 | 66.04 |
| 12 | Sodankyla | SOD | 67.37 | 26.63 | 63.96 | 63.96 |
| 13 | Meanook | MEA | 54.62 | 246.65 | 61.17 | 61.17 |
| 14 | Macquarie Island | MCQ | -54.50 | 158.95 | -59.51 | 59.51 |
| 15 | Nurmijarvi | NUR | 60.51 | 24.66 | 57.79 | 57.79 |
| 16 | Port Aux Francais | PAF | -49.35 | 70.26 | -56.46 | 56.46 |
| 17 | Ottawa | OTT | 45.40 | 284.45 | 54.88 | 54.88 |
| 18 | Niemegk | NGK | 52.07 | 12.68 | 51.64 | 51.64 |
| 19 | Crozet | CZT | -46.43 | 51.87 | -51.06 | 51.06 |
| 20 | Furstenfeldbruck | FUR | 48.17 | 11.28 | 48.09 | 48.09 |
| 21 | Hurbanovo | HRB | 47.87 | 18.19 | 46.66 | 46.66 |
| 22 | Martin De Vivies | AMS | -37.80 | 77.57 | -45.83 | 45.83 |
| 23 | Tihany | THY | 46.90 | 17.89 | 45.76 | 45.76 |
| 24 | Gnangara | GNA | -31.78 | 115.95 | -41.18 | 41.18 |
| 25 | Learmonth | LRM | -22.22 | 114.10 | -31.70 | 31.70 |
| 26 | Charters Towers | CTA | -20.09 | 146.26 | -27.37 | 27.37 |
| 27 | Tamanrasset | TAM | 22.79 | 5.53 | 24.30 | 24.30 |
| 28 | Kakadu | KDU | -12.69 | 132.47 | -21.34 | 21.34 |
| 29 | M'Bour | MBO | 14.38 | 343.03 | 19.59 | 19.59 |
| 30 | Papeete | PPT | -17.57 | 210.43 | -15.04 | 15.04 |
| 31 | Kourou | KOU | 5.21 | 307.27 | 14.27 | 14.27 |
| 32 | Phu Thuy | PHU | 21.03 | 105.95 | 11.22 | 11.22 |
| 33 | Bangui | BNG | 4.33 | 18.57 | 4.04 | 4.04 |
| 34 | Ascension Island | ASC | -7.95 | 345.62 | -2.74 | 2.74 |

**Table 1**        The name, ABB code, geographic latitude, geographic longitude and geomagnetic latitude, and the absolute geomagnetic latitude of the analysed stations.



| Sl No. | Station | GM lat (Abs) | Year Avg VV GM(H) | | | | | | | | | | | | Avg VV GM(H) Solar cycle 23 |
|---|---|---|---|---|---|---|---|---|---|---|---|---|---|---|---|
| | | | 1997 | 1998 | 1999 | 2000 | 2001 | 2002 | 2003 | 2004 | 2005 | 2006 | 2007 | 2008 | |
| 1 | ALE | 87.38 | 157.73 | 219.61 | 241.07 | 260.96 | 262.52 | 245.41 | 367.27 | 247.68 | | | | | 250.28 |
| 2 | THL | 87.02 | | | 126.91 | 162.75 | 103.66 | 103.18 | 159.44 | 102.28 | 159.44 | 82.85 | 81.54 | 79.38 | 116.14 |
| 3 | RES | 82.61 | 96.35 | 118.24 | 134.10 | 146.22 | 137.81 | 136.76 | 213.27 | 139.74 | 155.29 | 119.94 | 120.18 | 118.84 | 136.39 |
| 4 | CBB | 76.21 | 151.89 | 179.02 | 203.41 | 224.57 | 190.37 | 195.37 | 353.04 | 232.57 | 253.84 | 210.20 | 207.88 | 197.83 | 216.67 |
| 5 | HRN | 74.02 | 231.42 | 259.38 | 277.98 | 292.59 | 246.34 | 278.66 | 485.73 | 344.47 | 372.26 | 287.32 | 305.32 | 303.24 | 307.06 |
| 6 | DRV | 73.87 | 140.38 | 135.89 | 151.20 | 177.67 | 152.10 | 236.26 | 257.40 | 276.14 | 185.98 | 273.21 | 177.23 | 131.10 | 191.21 |
| 7 | IQA | 73.25 | 277.97 | 300.81 | 330.56 | 350.91 | 285.13 | 324.94 | 577.90 | 413.27 | 422.51 | | 353.51 | 349.37 | 361.52 |
| 8 | MAW | 73.04 | | 304.11 | 313.69 | 321.06 | 287.86 | 294.60 | 530.68 | 358.46 | 393.54 | 290.32 | 298.48 | 291.09 | 334.90 |
| 9 | BLC | 72.66 | 224.50 | 254.94 | 278.18 | 293.60 | 249.75 | 269.56 | 476.25 | 337.46 | 376.19 | 296.56 | 289.32 | 294.72 | 303.57 |
| 10 | YKC | 68.62 | 256.86 | 297.90 | 331.45 | 340.99 | 275.45 | 309.94 | 558.26 | 375.28 | 393.88 | 285.04 | 278.33 | 279.65 | 333.03 |
| 11 | ABK | 66.04 | 192.58 | 248.54 | 271.75 | 299.37 | 244.47 | 269.20 | 494.21 | 293.66 | 307.73 | 205.57 | 186.16 | 169.91 | 265.26 |
| 12 | SOD | 63.96 | 137.42 | 190.07 | 205.53 | 238.06 | 192.54 | 209.55 | 391.39 | 221.98 | 231.38 | 149.55 | 129.90 | 117.16 | 201.21 |
| 13 | MEA | 61.17 | 120.98 | 173.65 | 181.81 | 225.08 | 176.65 | 183.51 | 339.59 | 339.70 | 207.60 | 118.69 | 96.96 | 91.29 | 187.96 |
| 14 | MCQ | 59.51 | | 197.39 | 204.30 | 236.53 | 198.53 | 207.24 | 396.88 | 213.76 | 234.39 | 140.72 | 120.69 | 112.52 | 205.72 |
| 15 | NUR | 57.79 | 42.41 | 57.78 | 58.33 | 81.66 | 73.85 | 66.75 | 115.18 | 67.26 | 66.26 | 43.50 | 39.96 | 38.10 | 62.59 |
| 16 | PAF | 56.46 | 41.56 | 63.16 | 62.59 | 91.77 | 86.76 | 77.66 | 134.50 | 81.52 | 74.96 | 57.53 | 47.34 | 41.89 | 71.77 |
| 17 | OTT | 54.88 | 40.22 | 55.38 | 54.38 | 70.47 | 60.65 | 52.62 | 89.58 | 56.31 | 55.70 | 35.66 | 34.65 | 40.94 | 53.88 |
| 18 | NGK | 51.64 | 41.67 | 36.74 | 35.29 | 44.22 | 37.86 | 37.03 | 56.17 | 31.91 | 34.01 | 25.22 | 24.73 | 23.79 | 35.72 |
| 19 | CZT | 51.06 | 21.31 | 26.00 | 28.49 | 35.60 | 34.54 | 28.70 | 43.92 | 27.20 | 23.07 | 20.79 | 16.42 | 17.28 | 26.94 |
| 20 | FUR | 48.09 | 21.79 | 25.37 | 26.23 | 31.75 | 28.70 | 27.54 | 39.24 | 21.90 | 23.88 | 17.80 | 17.67 | 16.87 | 24.89 |
| 21 | HRB | 46.66 | | 24.36 | 28.93 | 34.58 | 31.11 | 29.28 | 42.57 | 24.23 | 26.49 | 19.45 | 18.97 | 18.24 | 27.11 |
| 22 | AMS | 45.83 | 21.99 | 24.77 | 27.33 | 31.50 | 28.36 | 29.33 | 39.48 | 26.98 | 22.26 | 17.99 | 23.36 | 18.59 | 26.00 |
| 23 | THY | 45.76 | | 28.51 | 29.23 | 35.24 | 31.78 | 30.17 | 43.24 | 24.57 | 26.52 | 19.49 | 19.01 | 18.31 | 27.82 |
| 24 | GNA | 41.18 | | 28.01 | 24.44 | 28.48 | 27.39 | 25.35 | 33.36 | 20.24 | 21.05 | 15.97 | 15.79 | 14.93 | 23.18 |
| 25 | LRM | 31.70 | 27.62 | 30.78 | 26.63 | 30.17 | 28.98 | 26.73 | 35.29 | 20.79 | 22.25 | 16.95 | 15.84 | 15.32 | 24.78 |
| 26 | CTA | 27.37 | | 27.89 | 22.73 | 27.91 | 25.84 | 22.82 | 30.38 | 16.87 | 18.84 | 13.95 | 13.81 | 13.09 | 21.28 |
| 27 | TAM | 24.30 | 19.45 | 32.75 | 23.05 | 30.57 | 25.62 | 24.50 | 31.44 | 18.01 | 22.29 | 14.22 | 14.50 | 13.59 | 22.50 |
| 28 | KDU | 21.34 | 17.82 | 19.12 | 21.23 | 24.32 | 22.29 | 20.31 | 25.67 | 13.82 | 15.24 | 11.11 | 10.90 | 10.33 | 17.65 |
| 29 | MBO | 19.59 | 19.11 | 22.34 | 23.92 | 29.60 | 26.78 | 25.75 | 31.33 | 18.16 | 19.62 | 14.69 | 14.77 | 15.80 | 21.82 |
| 30 | PPT | 15.04 | 30.58 | 28.61 | | 60.79 | 83.68 | 22.78 | 23.70 | 14.10 | 13.66 | 9.79 | 11.48 | 10.24 | 28.13 |
| 31 | KOU | 14.27 | 19.87 | 37.46 | 29.91 | 26.51 | 30.46 | 22.45 | 24.77 | 14.85 | 14.85 | 12.29 | 11.01 | 9.93 | 21.07 |
| 32 | PHU | 11.22 | | 20.03 | 26.50 | 29.04 | 30.11 | 27.40 | 31.04 | 18.90 | 19.56 | 13.90 | 13.34 | 12.75 | 22.07 |
| 33 | BNG | 4.04 | 19.59 | 22.73 | 23.46 | 27.94 | 26.60 | 28.51 | 30.34 | | | 23.92 | 16.67 | 15.53 | 23.51 |
| 34 | ASC | 2.74 | 25.29 | 25.02 | 39.57 | 28.73 | | | 23.36 | 12.98 | 14.04 | 10.85 | 12.64 | 9.17 | 20.17 |

**Table 2**    The ABB code, absolute geomagnetic latitude, and the year and solar cycle (23) means of the VV values of

5    the analysed stations.



| Year | Post-Auroral Lat | | | Pre-Auroral Lat | | | | Mid Lat | | | | Low Lat | | |
|------|------|------|--------|------|------|------|--------|------|------|------|--------|------|------|--------|
| | RES | CBB | Avg VV | IQA | MAW | SOD | Avg VV | OTT | NGK | CZT | Avg VV | BNG | ASC | Avg VV |
| 1997 | 96.35 | 151.89 | 124.12 | 277.97 | | 137.42 | 207.70 | 40.22 | 41.67 | 21.31 | 34.40 | 19.59 | 25.29 | 22.44 |
| 1998 | 118.24 | 179.02 | 148.63 | 300.81 | 304.11 | 190.07 | 265.00 | 55.38 | 36.74 | 26.00 | 39.37 | 22.73 | 25.02 | 23.88 |
| 1999 | 134.10 | 203.41 | 168.75 | 330.56 | 313.69 | 205.53 | 283.26 | 54.38 | 35.29 | 28.49 | 39.39 | 23.46 | 39.57 | 31.51 |
| 2000 | 146.22 | 224.57 | 185.40 | 350.91 | 321.06 | 238.06 | 303.34 | 70.47 | 44.22 | 35.60 | 50.10 | 27.94 | 28.73 | 28.34 |
| 2001 | 137.81 | 190.37 | 164.09 | 285.13 | 287.86 | 192.54 | 255.17 | 60.65 | 37.86 | 34.54 | 44.35 | 26.60 | | 26.60 |
| 2002 | 136.76 | 195.37 | 166.07 | 324.94 | 294.60 | 209.55 | 276.36 | 52.62 | 37.03 | 28.70 | 39.45 | 28.51 | | 28.51 |
| 2003 | 213.27 | 353.04 | 283.16 | 577.90 | 530.68 | 391.39 | 499.99 | 89.58 | 56.17 | 43.92 | 63.22 | 30.34 | 23.36 | 26.85 |
| 2004 | 139.74 | 232.57 | 186.15 | 413.27 | 358.46 | 221.98 | 331.23 | 56.31 | 31.91 | 27.20 | 38.47 | | 12.98 | 12.98 |
| 2005 | 155.29 | 253.84 | 204.57 | 422.51 | 393.54 | 231.38 | 349.14 | 55.70 | 34.01 | 23.07 | 37.59 | 23.92 | 14.04 | 18.98 |
| 2006 | 119.94 | 210.20 | 165.07 | | 290.32 | 149.55 | 219.93 | 35.66 | 25.22 | 20.79 | 27.22 | 16.67 | 10.85 | 13.76 |
| 2007 | 120.18 | 207.88 | 164.03 | 353.51 | 298.48 | 129.90 | 260.63 | 34.65 | 24.73 | 16.42 | 25.27 | 15.53 | 12.64 | 14.09 |
| 2008 | 118.84 | 197.83 | 158.33 | 349.37 | 291.09 | 117.16 | 252.54 | 40.94 | 23.79 | 17.28 | 27.33 | | 9.17 | 9.17 |

**Table 3**    The yearly means of VV values of selected stations from 1997 through 2008 and their latitude category

5 VV averages.

| Year | SW speed (Km/s) |
|------|-----------------|
| 1997 | 379.71 |
| 1998 | 412.69 |
| 1999 | 441.36 |
| 2000 | 446.36 |
| 2001 | 426.00 |
| 2002 | 441.69 |
| 2003 | 545.43 |
| 2004 | 446.92 |
| 2005 | 466.93 |
| 2006 | 434.54 |
| 2007 | 448.47 |
| 2008 | 448.62 |

10 **Table 4**    The year averages of the solar wind velocity for the years from 1997 through 2008.



| Avg VV GM(H)-SW Correlation | |
| --- | --- |
| **Station Category** | **Correlation** |
| Post-Auroral  Lat | 0.96 |
| Pre-Auroral Lat | 0.91 |
| Mid Lat | 0.59 |
| Low Lat | 0.06 |

**Table 5**          The correlation coefficients obtained for average VV values of the post-auroral, pre-auroral, mid-latitude

and low-latitude station groups with the solar wind velocity during the years 1997-2008.





# Figures

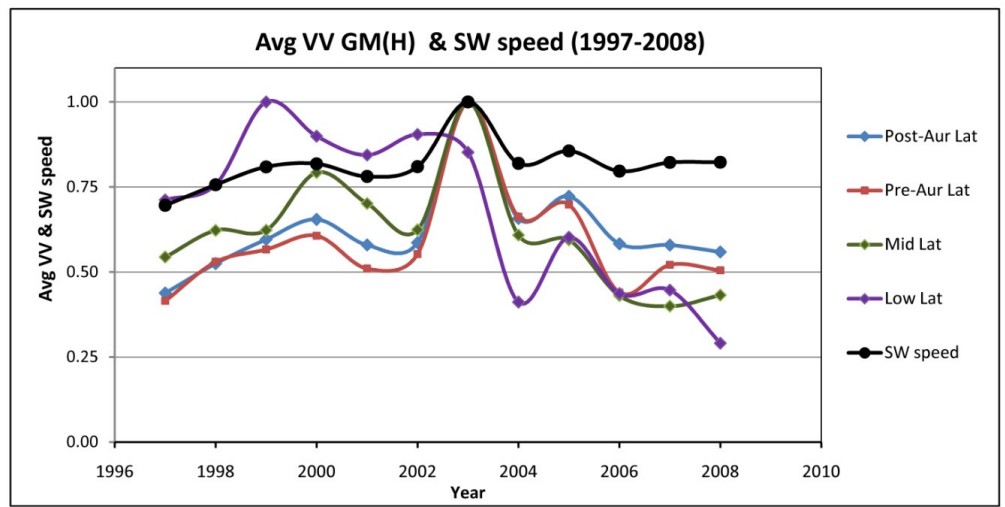

**Figure 1**          Normalised station category averages of yearly VV values and solar wind speed for the years from 1997

through 2008.



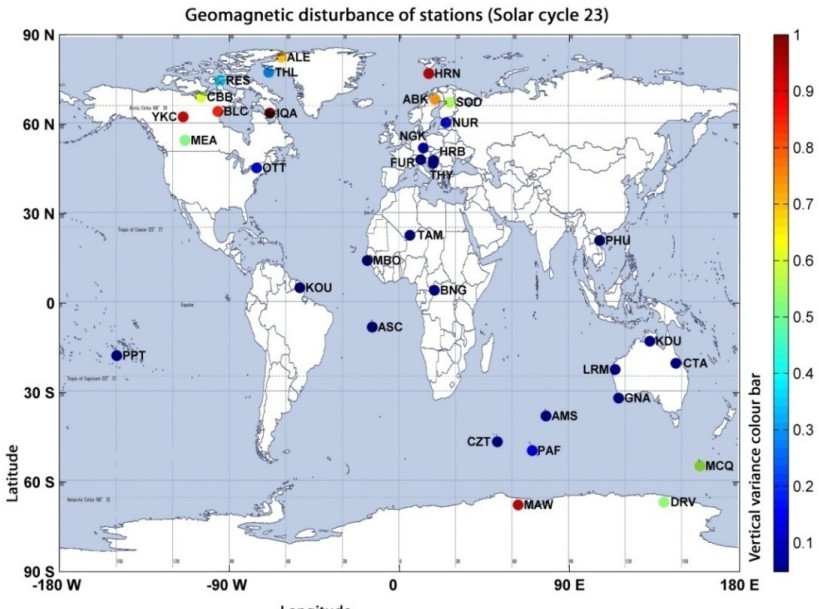

**Figure 2**        The disturbance markers for the analysed GM stations based on the mean VV values for solar cycle 23.




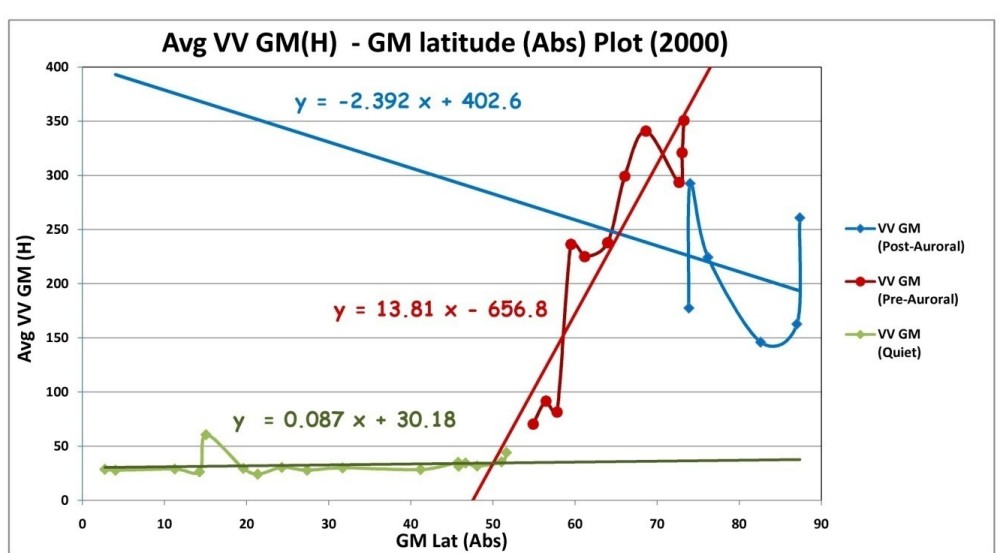

**Figure 3**      The plot of the average of VV values and the best fits of the analysed stations for year 2000 against absolute GM latitudes.




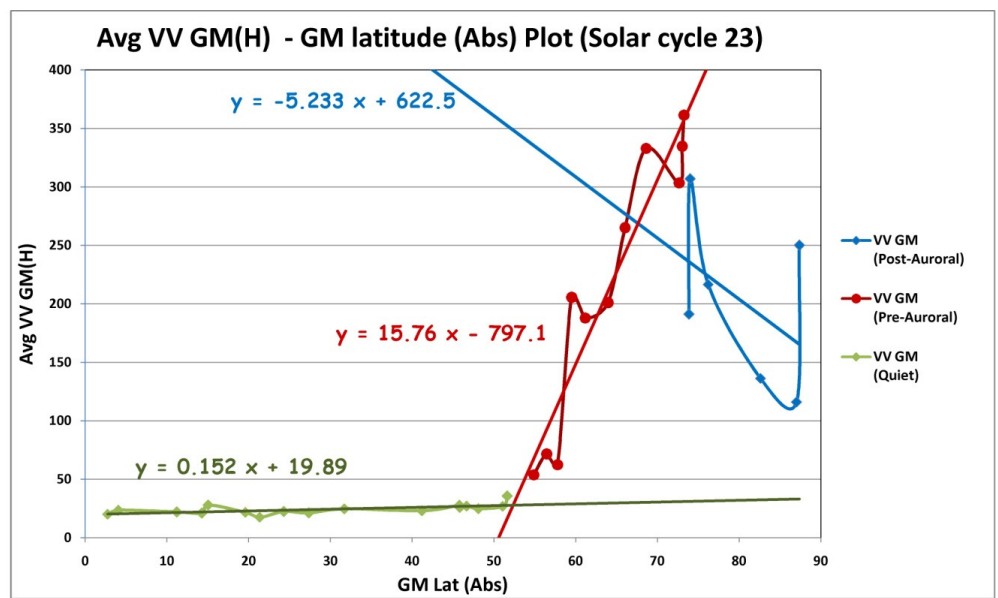

**Figure 4**      The plot of the average of VV values and the best fits of the analysed stations for solar cycle 23 against absolute GM latitudes.