# Peer review of "Singular variations in geomagnetic disturbance content at auroral latitudes"

_Annales Geophysicae, 2018_

## Referee Comment (RC1) · Anonymous Referee #1 · 3 Aug 2018

This manuscript has serious problems in reaching the scientific level required of publications in AnnGeo. The manuscript contains no important new scientific results, and mainly repeats already known facts of geomagnetic activity. The main "results" of the manuscript listed in the abstract are that the "disturbance content is low for equatorial stations and high for auroral stations", with latitude profile exhibiting " 'Knee behaviour'.. beyond critical latitude" (of about 52 deg), and that this increasing pattern "ends around the auroral oval". All these facts are known since, at least, 70 years. The manuscript is overall rather ignorant of the scientific status in the field. It gives no proper background or relevant introduction for the study, uses rather trivial and partly unreliable methods, and includes references to articles that are partly irrelevant. It is clear that this manuscript must be completely revised and cannot be published in anything that

resembles the current version. Alas, the authors have collected a database which may be used to develop something new. However, the current version is not sufficient for a scientific publication. I list more details below to support my statement.

The increase of geomagnetic activity ("disturbance content") with latitude to a maximum in the auroral region, and the fact that this increase is not steady but experiences a rather sudden increase at about 50-55deg was already known, e.g., to J. Bartels (probably even by earlier researchers) who developed the normalization method for stations at different latitudes to obtain the multi-station planetary Kp/Ap index in 1930s. More recently, these results have been shown and quantified very clearly, e.g., by Svalgaard and Cliver (JGR, 2007; Fig.5). This figure also shows the decrease of activity beyond the oval.

Introduction is largely out of focus. Authors are mainly discussing the Earth's main magnetic field and its (long-term) change, which is not the topic of their manuscript, rather than geomagnetic activity and its solar wind drivers (see also below). This shows ignorance of the proper context.

Authors introduce a new daily parameter of geomagnetic activity, which is a kind of minute-scale version of the hourly (full-day) IHV index (Svalgaard et al., 2004). However, it is not clear what benefits of any, this parameter may offer. Anyway, the presented results based on the new parameter show nothing essentially new.

The method of determining the "knee" latitude is rather trivial and even arbitrary. It depends on the pre-chosen division of stations. Moreover, it seems that at least one of the stations (PPT) seems to have some problems, which affects the green line slope to be smaller in Fig. 3. This reduces the knee latitude of that case. Anyway, the results are not convincing or objective, and do not improve the earlier estimates of the knee latitude.

Discussion of results shows unawareness of basic structures of the magnetosphere and of solar wind-magnetosphere interaction. Authors seem to believe that the main

cause of geomagnetic activity is IMF strength, and that the auroral oval has activity maximum because more IMF is concentrated there. Because these erroneous views, they have hard time in explaining the lower activity in the polar cap (poleward of the oval) which is even more directly connected to the IMF than the oval. Even some references like Vokhmyanin and Ponyavin (2012) are misplaced. This paper discusses IMF polarity effect, not geomagnetic activity.

The authors mention the "associated high geo-electric hazard which induce damages in technological systems" several times. Such hazards are also known since several decades. Nothing new about such hazards are included in this manuscript.

Presentation of material is rather elementary and occasionally somewhat tedious to read. Authors repeat some topics several times. Statements like "Solar wind is hence established as the primary driver of geomagnetic disturbance" and "..figures point to a coupling between the IMF and GM field, strongest at high latitudes and weakening towards the low latitudes " show hyperbole reflecting ignorance.

It is not clear to me what they call "singular variations". They do not explicitly explain this nor, e.g., use singular component or principal component analysis methods. This term floats in the air. Also term vertical variation (VV) is inappropriate, since variations in the H component are mainly due to horizontal, not vertical currents.

---

## Referee Comment (RC2) · Anonymous Referee #2 · 7 Aug 2018

**1. Introduction**

The authors justify the motivation for their work on the basis that "geomagnetic disturbances induce currents in technological systems such as power grids . . .. etc". That is correct but is not relevant to the analysis conducted in this paper. The analysis involves short-term (one-minute) fluctuations in the earth's magnetic field. These small variations do not induce significant currents in long power lines, or long pipelines. Large DC currents in the auroral electrojet and equatorial electrojet induce ground currents, and currents in long conductors on the ground (eg oil pipelines in Nigeria and Alaska, long power-lines under the auroral electrojet current. Fluctuations in B that are relatively small magnitude are not significant. My point is that the calculated values are not relevant to the stated goal. There may be another application but it is not clear what

this might be.

2. Methodology and 3. Analysis.

There is nothing intrinsically incorrect with the methodology in the calculation of the authors' "VV" index. It simply involves using successive differences between magnetic field values each minute. Presumably the one minute time step was chosen because that was available from the data-base. It is not clear to me how this VV index relates to physical processes.

3.1 Mean Value Analysis and Validation and 3.2 Spatial disturbance marking Line 4 page 5: Should "aa index" be "AE index" ?

Auroral zone stations experience, on average, greater magnetic disturbances (DC and AC variations). The auroral current systems moves to lower latitudes in response to solar activity and changes in the Bz component of the IMF field. Thus it is not surprising that mid-latitude stations also experience magnetic field variations. Contrary to the statement given by the authors, the solar wind speed is not the prime driver of geomagnetic disturbances. The solar wind speed is one of many variables that affect observations. It is also not surprising that equatorial stations show minimal 1-minute fluctuations.

3.3 Threshold latitude and 4. Discussion

The 50 degree latitude, above which, B fluctuations are bigger than lower latitudes. This is addressed above. The authors suggest the 'knee' latitude provides insight into why there is an "increased geo-electric risk" at higher latitudes. This is an erroneous conclusion because the short-term B variations do not induce currents in ground conductors. It is the large DC currents that induce currents. The long oil pipeline in Nigeria has large induced currents due to the equatorial electrojet, but as the authors have pointed out, there is minimal B fluctuation at these low latitudes.

Other Comments

Table 1. There are two columns labels "GG". Should be "GG" and "GL" ??

Title: What does the word "Singular" mean in the context of this paper?

Page 5 line 17: Pearson is not referenced.

IN SUMMARY: I can not recommend publication and I am unable to see how minor or major changes might make it acceptable.

There is no explanation why this VV index has any merits as compared to other magnetic indices. The VV index is not related to any "geo-electric" effects suggested by the authors.